# Pancreatic Solid Pseudopapillary Neoplasms—Clinicopathological Characteristics and Surgical Outcomes: A 10-Year Single-Centre Observational Study

**DOI:** 10.3390/biomedicines13092050

**Published:** 2025-08-22

**Authors:** Agnieszka Partyka, Wiktoria Bajek, Paulina Wietrzycka, Beata Jabłońska, Sławomir Mrowiec

**Affiliations:** 1Student Scientific Society, Department of Digestive Tract Surgery, Faculty of Medical Sciences in Katowice, Medical University of Silesia, 14 Medyków Street, 40-752 Katowice, Poland; agnieszka.partyka23@gmail.com (A.P.); victoriabajek@gmail.com (W.B.); ppwietrzycka@gmail.com (P.W.); 2Department of Digestive Tract Surgery, Faculty of Medical Sciences in Katowice, Medical University of Silesia, 14 Medyków Street, 40-752 Katowice, Poland; mrowasm@poczta.onet.pl

**Keywords:** solid pseudopapillary neoplasm, pancreas, surgery

## Abstract

**Background**: Pancreatic solid pseudopapillary neoplasms (SPNs) are rare exocrine tumours with predominance in young women. These tumours are of low malignant potential, become considerably large before causing symptoms and are associated with good prognosis. This study aimed to present and analyse clinicopathological features and surgical outcome of SPNs. **Methods**: A retrospective analysis of 22 patients who underwent pancreatic surgery for SPNs in a single high-volume surgical centre in 2014–2023 was performed. **Results**: SPN was the most frequent in females (n = 21, 95.45%) in a mean age of 34 ± 11.09 (18–55) years. Fourteen (63.64%) patients were asymptomatic, and eight (36.36%) presented with symptoms. The most common clinical symptom was abdominal pain (n = 7, 31.82%). The majority of tumours were located in the pancreatic body (n = 8, 36.36%), and most patients underwent distal pancreatectomy (n = 11, 50%). The median tumour size was 3.6 cm (IQR = 4.9; range: 1.3–14). The median duration of hospitalisation was 12.5 days, and the postoperative complication rate was 40.91%. R0 resection was achieved in 18 (81.82%) patients. Postpancreatectomy acute pancreatitis (PPAP) was the most common postoperative complication. No adjuvant therapy in any patient was needed. One-year overall survival (OS) equalled 100% and five-year OS reached 85%. None of the patients developed diabetes or signs of impaired pancreatic secretion in the follow-up period. Histopathology showed features like perineural invasion in 72.73% of cases, pseudocapsule (59.09%), haemorrhage (45.45%), vascular invasion (40.91%), mucosal metaplasia (40.91%), necrosis (31.82%), and calcification in the capsule (31.82%). Ki67 did not exceed 7%. In one case (4.55%), metastasis to a lymph node was found. Clinical suspicion agreed with histopathological results in only 10 (45.45%) cases. **Conclusions**: SPN most often occurs in young females. The majority of cases are asymptomatic accidental findings. The final diagnosis of SPN can be based just on analysis of histopathological examination results.

## 1. Introduction

Solid pseudopapillary neoplasms of the pancreas (SPNs), also known as Franz tumours, Hamoudie tumours, solid–cystic–papillary epithelial neoplasms and solid and cystic tumours, were designated as SPNs by the WHO in 2010 [1]. SPNs are of low malignant potential, become considerably large before causing symptoms and are associated with good prognosis [2,3,4].

SPNs constitute approximately 1–3% of all exocrine pancreatic tumours [2,5,6]. The female gender is an important predisposing factor. SPNs are also infrequent in males [7,8,9], and mainly affect young women. The average morbidity of this disease occurs in the third and fourth decade of life [10,11,12,13,14].

The pathogenesis is unclear, but there are reasons to associated the development of the tumour with oestrogen [15]. Gene mutations and tumour stroma are different between genders [16].

There are nonspecific clinical presentations [5]. The common symptom is related to nonspecific abdominal pain [11]. Other uncharacteristic clinical symptoms are nausea, vomiting, fever, weight loss and jaundice [5].

Detection of these tumours has increased in the last decade, which is connected in part to the increased use of imaging investigating different abdominal symptoms, whilst others are accidental discoveries [5,17].

SPNs are rare tumours that consist of weakly composite epithelial cells which originate from pancreatic pluripotent stem cells [2,18]. SPNs typically undergo cystic degeneration [19].

In larger tumours, solid and cystic areas are distinguished [19]. Microscopically, a pseudopapillary appearance is visible, resulting from degenerative changes [19]. This neoplasm shows some specific morphologic features, such as infiltration of the pancreatic parenchyma, capsule invasion, perineural invasion and angiovascular invasion, although the differential diagnosis with other pancreatic neoplasms that are neuroendocrine tumours may be a difficult task, especially without biopsy tests or histopathology results and only with imaging tests [20].

SPNs are frequently located in the pancreatic head or tail and are often misdiagnosed [8].

Chemotherapy and radiotherapy have shown little or no advantage in the treatment of SPNs [3]. The primary treatment is surgery [3]. There are several possible types of surgery, including those related to the location of the tumour [21].

The aim of this study was a retrospective analysis of the clinicopathological characteristics and surgical outcomes of patients who underwent surgery for SPN.

## 2. Materials and Methods

### 2.1. Patients

The analysis included 22 patients with SPN who underwent pancreatic surgery in the Department of Digestive Tract Surgery, Medical University of Silesia, Katowice, Poland, in the period 2014–2023. The following inclusion criteria were used: primary SPN, age > 18 years, surgical treatment and histopathological confirmation. Exclusion criteria included recurrent SPN and incomplete demographic and/or clinical data. All procedures performed in studies involving human participants were in accordance with the 1964 Helsinki Declaration and its later amendments or comparable ethical standards. General patient characteristics are presented in Table 1.

### 2.2. Study Design

This was a retrospective study of medical records and all data were fully anonymised. All procedures performed in studies involving human participants were in accordance with the 1964 Helsinki Declaration and its later amendments or comparable ethical standards. According to national and local legislation, only a medical experiment needs to obtain ethical approval. This retrospective study including analysis of patients’ medical records did not need this approval.

Medical records were retrospectively reviewed. Clinical and pathological features, including age, gender, body mass index, clinical symptoms, radiological reports, duration of hospitalisation, type and duration of surgery, tumour location and size, clinical and histopathological staging and grading, postoperative complications and overall survival, were collected and analysed. Complications were graded using the Clavien–Dindo classification (Dindo et al., 2004) [22]. Severe complications were defined as Clavien–Dindo grade III or greater. Postoperative complications were defined according to the International Study Group of Pancreatic Surgery (ISGPS). Follow-up time was defined as the interval between the date of first surgery and the date of last follow-up or imaging in an outpatient clinic.

### 2.3. Statistical Analysis

The categorical variables were presented as numbers and percentages and compared using the Pearson *χ*^2^ test. Continuous variables with normal distribution were expressed as means and standard deviation and analysed using Student’s *t* test for data. Continuous variables with non-normal distribution were shown as the median and interquartile range and analysed using the Mann–Whitney test. The Shapiro–Wilk test was used to determine statistical distribution in the analysed patients. Prevalence and frequency were expressed as number and percentage. A *p* < 0.05 was considered to be statistically significant. The statistical analyses were performed using TIBCO Statistica^®^ 13.3.

## 3. Results

### 3.1. Patients’ General Characteristics

There were 21 women and 1 man in the analysed group, with a mean age of 34 ± 11.09 years (range: 18–55 years). The clinicopathological features of the 22 patients are listed below (Table 2). Median body mass index (BMI) equalled 26.1 (18.2–40.23; IQR: 8.17). Most of the patients were overweight or obese.

### 3.2. Clinical Symptoms

A total of 8 out of 22 patients presented with symptoms, but 14 were asymptomatic. Clinical presentation was nonspecific regarding the symptomatic patients. Abdominal pain was the most frequent symptom (n = 7). Others occurred as follows: abdominal discomfort (n = 3), abdominal distension (n = 2), constipation (n = 2), loss of body weight (n = 2), nausea (n = 1), vomiting (n = 1) and asthenia (n = 1). None of the patients had jaundice, fever, diarrhoea, anaemia, cachexia or history of trauma.

Sixteen patients had comorbidities.

Eight patients had previous abdominal surgeries. Four declared significant cigarette smoking. Four patients had family history of malignant neoplasms.

### 3.3. Preoperative Diagnostic Imaging

All of the patients had received computed tomography (CT) and ultrasound (USG), and seven patients had undergone magnetic resonance imaging (MRI). Lesions were localised in the pancreatic head (n = 7), neck (n = 2), body (n = 8) and tail (n = 5). Endoscopic ultrasound (EUS) was performed in 11 cases, with fine-needle aspiration (FNA) in 8 patients. Specific data are presented in Table 3.

### 3.4. Preoperative Diagnosis

None of the patients had a definitive preoperative diagnosis, and a proper potential diagnosis was made for 10 patients. Commonly incorrect diagnoses included neuroendocrine tumours (NET) (n = 6), gastrointestinal stromal tumours (GIST) (n = 3), serous cystadenomas (n = 1), pancreatoblastoma (n = 1) and tropical calcific pancreatitis (TCP) (n = 1).

### 3.5. Surgeries

All patients underwent various surgical procedures depending on the location of the tumour. A total of 20 patients had pancreatectomies, including 6 pancreatoduodenectomies (PD) with the Traverso–Longmire technique, 3 central pancreatectomies (CP) and 11 distal pancreatectomies (DP) (including 5 spleen-preserving resections). One patient underwent enucleation of the tumour located in the pancreatic uncinate process and one patient had duodenum-preserving pancreatic head resection (Table 4).

Median duration of surgery was 312.5 (140–570) min. The majority of the patients (n = 16) were categorised as group 2 in the ASA classification (Table 4). The resection margin was negative (R0) in 18 cases and positive in 4 cases (R1: n = 3 and R2: n = 1).

### 3.6. Postoperative Complications

Postoperative complications occurred in nine patients and were categorised using the Clavien–Dindo classification. The most common ones were postpancreatectomy acute pancreatitis (PPAP) (n = 3), clinically relevant (CR) postoperative pancreatic fistulas (POPF) [Grade B (3), and C (2)], hydrothorax (2) and pneumothorax (1) (Table 5).

All patients with PPAP and hydrothorax were treated conservatively with positive outcomes. In two cases of POPF, grade B prolonged drainage was needed. Two patients required relaparotomy with irrigation and drainage because of grade C POPF. In one patient a successful pneumothorax drainage was performed. One patient had to be rehospitalised 10 days after the surgery due to PPAP but the mentioned conservative treatment and a course of antibiotics led to full recovery. There were no surgical mortalities within 30 days after the procedures.

### 3.7. Histopathological Results

#### 3.7.1. Histopathological Features

Median tumour size was 3.6 cm with interquartile range (IQR) of 4.9 (range: 1.3–14). Histopathology results showed features like perineural invasion, pseudocapsule, resorption, haemorrhage, vascular invasion, mucosal metaplasia, hyalinosis, infiltration of the pancreatic parenchyma, necrosis, calcification of the capsule and lobular metaplasia, but only the number of cases with infiltration of the pancreatic parenchyma was significantly higher in body and tail tumours (Table 6). Median marker of proliferation Ki-67 equalled 3% (range: 1–7%; IQR: 1).

Taking into consideration the presence of features of malignancy such as vascular and perineural invasion or infiltration of the pancreatic parenchyma, 20 tumours were malignant and 2 neoplasms were benign.

Thirteen tumours were classified as the pT3N0 group, five as the pT2N0 group and three cases were categorised as the pT1N0 group in the AJCC staging. In one patient, metastasis to a lymph node was found, which determined the pT3N1 group in the histopathology results.

#### 3.7.2. Immunohistochemistry

Immunohistochemical staining was performed on all neoplasms. The analyses typically showed positive findings when stained for beta-catenin, synaptophysin, CD10, cytokeratin AE1/AE3, PAS, CD56, SOX11 and vimentin. Tumours were stained negative for desmin, E-cadherin, neurofilament and BCL10. Progesterone receptors were present in three cases. Chromogranin A, neuron-specific enolase, CD68, p53, CD117, S100 and cyclin D1 were expressed variably in few cases (Table 6).

### 3.8. Follow-Up

No postsurgical adjuvant therapy was administered to any patient. One-year overall survival for 16 patients equalled 100%, with six lost to follow-up. Five-year overall survival reached 85%. One month after the surgery, a pancreatic cyst was found in one patient. Another patient developed a wound abscess 2 months after the procedure. Both cases were treated conservatively. One patient developed local recurrence 10 months after pancreatoduodenectomy and had to be reoperated on, which led to total pancreatectomy. The procedure was successful with no complications. In two cases, removal of the drain 2 and 3 months after initial surgeries led to brief rehospitalisations. Three years after the first surgery, one patient was diagnosed with chronic pancreatitis and abscess that led to relaparotomy and distal pancreatectomy due to the risk of necrosis of anastomosis. There were no postoperative complications. During the follow-up period, none of the patients developed diabetes or signs of impaired pancreatic secretion.

## 4. Discussion

SPN is a rare primary tumour of the pancreas, considered to be low-grade-malignant, and represents only about 1% of all tumours of the pancreas, but its diagnosis has been increasing in the last decade. The origin of SPN is unclear. There are two main theories. The first one says that SPN originates from multipotent primordial cells. The other one suggests an extrapancreatic origin, from genital ridge angled cells. The recurrence of the tumour is possible after surgery, so routine follow-up with imaging is mandatory. In the conducted small group study, only 1 of 22 patients experienced recurrence and needed another operation. SPNs most often metastasise to the liver, regional lymph nodes, mesentery, omentum and peritoneum; however, none of the patients had metastases to the mentioned organs [23]. In general, our findings are similar to the ones in the literature, although it is important to highlight that SPNs are rare tumours and the presentation of them can differ in each person [8,24,25,26].

Many studies show a predominance of SPN in young women, which is seen in our study as well. The mean age of our patients was 34 years, and according to the literature it is the most common in the third and fourth decade of life [10,11,12,13,14,24]. The most common location for SPN is the body and tail, and then the head and neck, of the pancreas [24,26].

Most patients with SPNs are asymptomatic. In our study, more than half of the patients had not experienced any kind of symptoms, which indicates the fact that diagnosis is often made incidentally during regular abdominal examination [7,8,11,24]. SPNs are often misdiagnosed because of their uncharacteristic features. In cases where symptoms were noticed, the most common ones were abdominal pain, abdominal discomfort and abdominal distention, exactly like in our study [7,8,24]. All of the patients had USG and multiphasic enhanced CT and, according to Mazzarella G. et al., this is the diagnostic test of choice [11]. In half of them (50%), EUS was performed, and 7 out of 22 (31.82%) received magnetic resonance imaging (MRI). According to European guidelines, EUS-FNA improves diagnostic accuracy in pancreatic cystic neoplasms (PCNs) for differentiating mucinous versus non-mucinous PCN, and malignant versus benign PCN, in cases where CT or MRI is unclear. However, FNB is considered a gold standard for the diagnosis of solid pancreatic lesions including SPN before surgery because obtaining a histologic core tissue specimen while using a biopsy needle and the wet-suction technique resulted in higher acquisition rate in comparison to the slow-pull method, with similar diagnostic accuracy between those two methods [27,28]. In every case mentioned, radiological tests confirmed the presence of masses arising from the pancreas. None of the patients had a definitive preoperative diagnosis, and a proper potential diagnosis was made for 10 patients (45.45%). Commonly incorrect diagnoses included neuroendocrine tumours (NETs), gastrointestinal stromal tumours, serous cystadenomas, pancreatoblastoma and tropical calcific pancreatitis. Papavramidis T and Papavramidis S suggest that the diagnostic algorithm should consist of USG, CT and MRI to provide information on the resectability of a tumour and CT-guided fine-needle aspiration because of some patients in which they helped in making a diagnosis [3].

SPNs can resemble neuroendocrine tumours due to their histomorphology and immunophenotype, so they have to be identified carefully [23,25]. Therefore fine-needle aspiration biopsy (FNA) may be useful in obtaining a diagnosis [11,23]. Histopathology results showed features like perineural invasion, pseudocapsule, resorption, haemorrhage, vascular invasion, mucosal metaplasia, hyalinosis, infiltration of the pancreatic parenchyma, necrosis, calcification of the capsule and lobular metaplasia which led to the diagnosis of pancreatic SPN [4,15,24].

The median size of tumours was 3.6 cm with interquartile range (IQR) of 4.9, while in other previous studies it differs from 5.4 to 9.5 cm [7,8,24].

The perfect treatment is accomplished by complete R0 resection of the tumour, preserving as much pancreatic tissue and other organs as possible. In our study, 18 patients had a negative resection margin (R0) and only 4 had a positive margin (R1 or R2). Surgical procedures were performed in all of the patients, and the type of surgery depended on the location of the tumour. Distal pancreatectomy can be performed for tumours in the body or tail of the pancreas, and pancreaticoduodenectomy for tumours of the pancreatic head [26,29,30,31,32]. Surgically resected SPNs are characterised by good overall survival [24]. The thirty-day mortality rate after surgeries in our group equalled 0. In unresectable SPNs, due to their size or widespread metastasis, radiotherapy is an option, as the tumour is radiosensitive [3]. In this study, none of the patients needed radiotherapy or adjuvant therapy. In our study, only nine patients had experienced any kind of postoperative complications. The most common one was postpancreatectomy acute pancreatitis (PPAP), but all the patients were treated conservatively with great outcomes. The other issues were postoperative pancreatic fistulas—grade B and C. Hydrothorax and pneumothorax occurred in one case each. There were no other complications in our group of patients. All of them were treated with good results and no surgical mortalities within 30 days of the procedures were seen. After median follow-up of 39.5 months, no recurrence of disease was shown in 16 out of 22 patients; the other 6 patients were lost to follow-up and their clinical status is unknown.

In compliance with other reports, our study demonstrates that SPNs rarely have lymph node metastases; therefore, extensive lymphadenectomy is not necessary [7,8]. Even though SPNs are malignant neoplasms, they are marked by indolent clinical courses. It is important to identify risk factors for malignancy, but for now there are still no criteria for defining malignant histopathological features.

In general, the prognosis of SPNs is good, even if the patients experience recurrence or have metastases. Most cases are limited to the pancreas and can be resected with a great result. In our group, only one patient needed to be reoperated on because of recurrence, but the second surgery went well with no complications. In our group, 1-year overall survival rate reached 100% and none of the patients developed diabetes or signs of impaired pancreatic secretion; however, according to the literature, the overall 5-year survival rate is around 95–98.8% [3,11,17,33]. Salvia et al. observed no recurrence after almost 60 months of follow-up [24].

This study has some limitations. First, it was a single-centre study with a group of 22 patients. Moreover it was a retrospective study of a rare condition. A prospective study should be conducted to validate the results in multiple centres.

## 5. Conclusions

SPN most often occurs in young females. The majority of cases are asymptomatic incidental findings. The final diagnosis of SNP can be based just on an analysis of histopathological examination results. The prognosis of SPN is great even if the metastases are present, and the surgical resection is recommended in this case as well. Overall survival ratio is around 95 to 98.8% with satisfying clinical status in patients.

## Figures and Tables

**Table 1 biomedicines-13-02050-t001:** General patient characteristics. Abbreviations: F—female, M—male, SPN—Solid pseudopapillary neoplasm, ASA PS CS—American Society of Anesthesiologists Physical Status Classification System, DP—distal pancreatectomy, CP—central pancreatectomy, PD—pandcreaticoduodenectomy, EUP—enucleation of the uncinate process, BP–Beger procedure, PPAP—acute postpancreatectomy pancreatitis, POPF—postoperative pancreatic fistulas.

Patient Number	Age at Operation (years)	Sex	BMI	Pathology	Presenting Symptoms	ASA PS CS	Type of Operation	Procedure	Tumour Location	Tumour Size (cm)	Complications	Length of Hospital Stay (days)	Recurrence	Adjuvant Treatment	Mortality	Follow-Up Time (years)
1	25	F	26.1	SPN	+	2	Laparotomy	DP	Tail	9.5	Hydrothorax	11	-	-	-	1
2	36	F	40.2	SPN	−	3	Laparotomy	DP	Tail	7	POPF Grade B	15	-	-	-	1.3
3	29	F	23.1	SPN	+	2	Laparotomy	DP	Body	2.5	-	7	-	-	-	3.3
4	18	F	25.1	SPN	−	2	Laparotomy	DP	Body	5	PPAP	7	-	-	-	3.3
5	45	F	32.8	SPN	−	2	Laparotomy	DP	Body	1.5	-	8	-	-	-	2.6
6	20	M	31.3	SPN	−	1	Laparotomy	PD	Head	8.5	PPAP	41	-	-	-	-
7	46	F	35.9	SPN	−	2	Laparotomy	EUP	Uncinate process	3.4	-	10	-	-	-	1
8	32	F	23.7	SPN	+	1	Laparotomy	DP	Tail	3.2	-	7	-	-	-	5.5
9	37	F	21.2	SPN	−	2	Laparotomy	CP	Neck	2.1	-	14	-	-	-	2.7
10	52	F	26.4	SPN	−	2	Laparotomy	DP	Body	5	-	10	-	-	-	5
11	23	F	31.7	SPN	+	2	Laparotomy	DP	Tail	10	POPF Grade B	12	-	-	-	3.2
12	44	F	20.1	SPN	+	1	Laparotomy	DP	Body	5	Pneumothorax	13	-	-	-	6
13	22	F	23	SPN	−	2	Laparotomy	BP	Body	1.3	Hydrothorax, POPF Grade B and C	59	-	-	-	5.5
14	41	F	31.1	SPN	+	2	Laparotomy	PD	Head	10.6	-	13	+	-	-	-
15	51	F	24	SPN	+	1	Laparotomy	CP	Body	1.7	-	18	-	-	-	1.3
16	25	F	31.7	SPN	−	2	Laparotomy	DP	Tail	14	POPF Grade C	29	-	-	-	5.3
17	31	F	19.8	SPN	+	2	Laparotomy	DP	Body	3	-	9	-	-	-	-
18	33	F	29.1	SPN	−	2	Laparotomy	CP	Neck	1.3	-	12	-	-	-	8.2
19	23	F	25.2	SPN	−	2	Laparotomy	PD	Head	3.8	-	10	-	-	-	7.8
20	26	F	30.1	SPN	−	1	Laparotomy	PD	Head	2	PPAP	30	-	-	-	-
21	55	F		SPN	−	2	Laparotomy	PD	head	2.5	-	20	-	-	-	-
22	34	F	18.2	SPN	−	2	Laparotomy	PD	Head	5.5	-	16	-	-	-	

**Table 2 biomedicines-13-02050-t002:** Patients’ characteristics. DM 1—type 1 diabetes mellitus; GERD—gastroesophageal reflux disease.

	Total (n = 22)	Head and Neck (n = 9)	Body and Tail (n = 13)	*p*-Value
**Age**	M: 34 ± 11.09 (18–55)	35 ± 11.3 (20–55)	33.3 ± 11.4 (18–52)	0.616
**Gender**				
Female	21 (95.45%)	8 (88.89%)	13 (100%)	0.218
Male	1 (4.55%)	1 (11.11%)	0	
**BMI**	M: 27.14 ± 5.71 (18.2–40.23) Me: 26.1 (IQR = 8.17)	M: 27.77 ± 5.85 (18.2–35.94) Me: 29.59 (IQR=8.03)	M: 26.75 ± 5.83 (19.75–40.23) Me: 25.08 (IQR = 8.53)	0.745
Underweight	1 (4.55%)	1 (11.11%)	0	
Normal	7 (31.82%)	1 (11.11%)	6 (46.15%)	
Overweight	5 (22.73%)	2 (22.22%)	3 (23.08%)	
Obesity	8 (36.36%)	4 (44.44%)	4 (30.77%)	
**Symptoms and signs**	8 (36.36%)	1 (11.11%)	7 (53.85%)	0.0405
Asymptomatic	14 (63.64%)	8 (88.89%)	6 (46.15%)	0.0404
Abdominal pain	7 (31.82%)	2 (22.22%)	5 (38.46%)	0.421
Abdominal discomfort	3 (13.64%)	0	3 (23.08%)	
Abdominal distension	2 (9.09%)	0	2 (15.38%)	
Constipation	2 (9.09%)	0	2 (15.38%)	
Loss of body weight	2 (9.09%)	0	2 (15.38%)	
Nausea	1 (4.55%)	0	1 (7.69%)	
Vomiting	1 (4.55%)	0	1 (7.69%)	
Asthenia	1 (4.55%)	0	1 (7.69%)	
Jaundice	0			
Fever	0			
Anaemia	0			
Cachexia	0			
History of trauma	0			
**Comorbidities**	16 (72.73%)	6 (66.67%)	10 (76.92%)	0.595
**Cardiac**				
Arterial hypertension	3 (13.64%)			
Mitral regurge	1 (4.55%)			
Varicose veins	1 (4.55%)			
**Respiratory**				
Asthma	2 (9.09%)			
Chronic sinusitis	1 (4.55%)			
**Neurological**				
Aneurysm IAS	1 (4.55%)			
**Gastrointestinal**				
Hypercholesterolemia	2 (9.09%)			
DM 1	1 (4.55%)			
GERD	1 (4.55%)			
Acute pancreatitis in history	1 (4.55%)			
Biliary gastropathy	1 (4.55%)			
Liver angiomas	1 (4.55%)			
Pancreatic cyst	1 (4.55%)			
Polyps of the gaster	1 (4.55%)			
**Renal**				
Nephrolithiasis	1 (4.55%)			
**Gynaecological**				
Endometriosis	1 (4.55%)			
Ovarian cysts	1 (4.55%)			
**Musculoskeletal**				
Rachialgia	3 (13.64%)			
Scoliosis	1 (4.55%)			
**Other**				
Depression	2 (9.09%)			
Hypothyroidism	2 (9.09%)			
Ehlers-Danlos syndrome	1 (4.55%)			

**Table 3 biomedicines-13-02050-t003:** Preoperative diagnostic imaging. ERCP—endoscopic retrograde cholangiopancreatography; FNA—fine-needle aspiration; PET—positron emission tomography.

Diagnostic Imaging Tests	Total (n = 22)	Head and Neck (n = 9)	Body and Tail (n = 13)	*p*-Value
Ultrasonography (USG)	22 (100%)	9 (100%)	13 (100%)	1.00
Computed tomography (CT)	22 (100%)	9 (100%)	13 (100%)	1.00
Magnetic resonance imaging (MRI)	7 (31.82%)	4 (44.44%)	3 (23.08%)	0.290
PET	2 (9.09%)	2 (22.22%)	0	
ERCP	1 (4.55%)	1 (11.11%)	0	
Endoscopic ultrasound (EUS)	11 (50%)	6 (66.67%)	5 (38.46%)	0.193
FNA of the pancreas	8 (36.36%)	4 (44.44%)	4 (30.77%)	0.512

**Table 4 biomedicines-13-02050-t004:** Surgeries and ASA classification. ASA—American Society of Anesthesiologists.

Procedure	Total (n = 22)	Head and Neck (n = 9)	Body and Tail (n = 13)	*p*-Value
Pancreatectomy	20 (90.91%)	8 (88.89%)	12 (92.31%)	0.952
Pancreatoduodenectomy (PD)	6 (27.27%)	6 (66.67%)	0	
Central (CP)	3 (13.64%)	2 (22.22%)	1 (7.69%)	0.399
Distal (DP)	11 (50%)	0	11 (84.62%)	
**Enucleation of the tumour** located in pancreatic uncinate process	1 (4.55%)	1 (11.11%)	0	
**Duodenum-preserving** **pancreatic head resection**	1 (4.55%)	0	1 (7.69%)	
**ASA classification**				
1	16 (72.73%)	2 (22.22%)	3 (23.08%)	0.688
2	5 (22.73%)	7 (77.78%)	9 (69.23%)	
3	1 (4.55%)	0	1 (7.69%)	

**Table 5 biomedicines-13-02050-t005:** Postoperative complications. PPAP—postpancreatectomy acute pancreatitis; CR-POPF—clinically relevant postoperative pancreatic fistula.

Complications	Total (n = 22)	Head and Neck (n = 9)	Body and Tail (n = 13)	*p*-Value
Clavien–Dindo classification				
Grade I	3 (13.64%)	1 (11.11%)	2 (15.38%)	0.801
Grade II	2 (9.09%)	1 (11.11%)	1 (7.69%)	0.803
Grade III	4 (18.18%)	0	4 (30.77%)	
PPAP	3 (13.64%)	2 (22.22%)	1 (7.69%)	0.399
CR-POPF				
Grade B	3 (13.64%)	0	3 (23.08%)	
Grade C	2 (9.09%)	0	2 (15.38%)	
Hydrothorax	2 (9.09%)	0	2 (15.38%)	
Pneumothorax	1 (4.55%)	0	1 (7.69%)	
Reoperation	2 (9.09%)		2 (15.38%)	
Rehospitalisation	1 (4.55%)	0	1 (7.69%)	
30-day mortality	0	0	0	

**Table 6 biomedicines-13-02050-t006:** Histopathological features and immunohistochemical staining.

Complications	Total (n = 22)	Head and Neck (n = 9)	Body and Tail (n = 13)	*p*-Value
Perineural invasion	16 (72.73%)	7 (77.78%)	9 (69.23%)	0.861
Pseudocapsule	13 (59.09%)	3 (33.33%)	10 (76.92%)	0.051
Infiltration of the pancreatic parenchyma	14 (63.64%)	3 (33.33%)	11 (84.62%)	0.018
Resorption	11 (50%)	4 (44.44%)	7 (53.85%)	0.801
Haemorrhage	10 (45.45%)	2 (22.22%)	8 (61.54%)	0.248
Vascular invasion	9 (40.91%)	3 (33.33%)	6 (46.15%)	0.694
Mucosal metaplasia	9 (40.91%)	2 (22.22%)	7 (53.85%)	0.324
Hyalinosis	8 (36.36%)	3 (33.33%)	5 (38.46%)	0.866
Necrosis	7 (31.82%)	3 (33.33%)	4 (30.77%)	0.927
Calcification of the capsule	7 (31.82%)	3 (33.33%)	4 (30.77%)	0.927
Lobular metaplasia	5 (22.73%)	1 (11.11%)	4 (30.77%)	0.382
Vascular embolism	2 (9.09%)	0	2 (15.38%)	
**Immunohistochemical Staining**				
Beta-catenin	21 (95.45%)	8 (88.89%)	13 (100%)	0.850
Synaptophysin	18 (81.82%)	7 (77.78%)	11 (84.62%)	0.897
CD10	15 (68.18%)	5 (55.56%)	10 (76.92%)	0.641
Cytokeratin AE1/AE3	9 (40.91%)	4 (44.44%)	5 (38.46%)	0.856
Periodic acid–Schiff (PAS)	7 (31.82%)	1 (11.11%)	6 (46.15%)	0.197
CD56	5 (22.73%)	3 (33.33%)	2 (15.38%)	0.438
SOX11	5 (22.73%)	1(11.11%)	4 30.77%)	0.382
Vimentin	4 (18.18%)	2 (22.22%)	2 (15.38%)	0.735
S100	3 (13.64%)	1 (11.11%)	2 (15.38%)	0.802
CD117	2 (9.09%)	0	2 (15.38%)	
p53	2 (9.09%)	1 (11.11%)	1 (7.69%)	0.803
Cyclin D1	2 (9.09%)	1 (11.11%)	1 (7.69%)	0.803
Chromogranin A	1 (4.55%)	0	1 (7.69%)	
Neuron-specific enolase (NSE)	1 (4.55%)	0	1 (7.69%)	
CD68	1 (4.55%)	1 (11.11%)	0	
Neurofilament	0	0	0	
BCL10	0	0	0	
Desmin	0	0	0	
E-cadherin	0	0	0	
HMB45	0	0	0	
Progesterone receptor	3 (13.64%)	1 (11.11%)	2 (15.38%)	0.802

## Data Availability

The original contributions presented in this study are included in the article. Further inquiries can be directed to the corresponding author.

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
