# Peer review of "Pancreatic Solid Pseudopapillary Neoplasms—Clinicopathological Characteristics and Surgical Outcomes: A 10-Year Single-Centre Observational Study"

_biomedicines, 2025, doi:10.3390/biomedicines13092050_

Round 1
Reviewer 1 Report
Comments and Suggestions for Authors
THe article is on an interesting and cutting edge topic. However, there are some limitations that should be addressed by the authors:
1) The study is retrospective and the sample size is limited. I am aware we are speaking about a rare disease but this is why multicenter series are needed in this field....
2) The KM is misleading as it appears as some patients died during the follow-up (they are just censored). I would remove it because it adds very little information to the paper
3) Why only half od patients underwent EUS-tisse acquisition before surgery? Don't you need a definitive diagnosis before surgery??
4) Why the patients underwent EUS-FNA instead of FNB which performs much better? In fact, most patients had incorrect or indefinite diagnosis......The authors should also comment in the Discussion that EUS-FNB is the gold standard for the diagnosis before surgery in these patients (in this regard, cite two relevant papers on the impact of FNB in solid pancreatic lesions including SPN: PMID: 36657607 and PMID: 35915956)
Author Response
Dear Reviewer,
This is the revised original article „Pancreatic solid pseudopapillary neoplasms – clinicopathological characteristics and surgical outcomes: a 10-year single-center observational study ” (biomedicines-3782585) for the special issue "Pancreatic, Liver, Biliary Tract and Intestinal Diseases: Pathogenesis, Diagnostics and Therapy—2nd Edition" of the “Biomedicines” journal.
Thank you for your questions and comments. We have fully addressed all the comments and our responses appear below. Our revised work includes corrections according to reviewers’ comments in the text. Our revisions, made according to reviewers’ comments, are marked using the red font function in the manuscript.
We take this opportunity to express our gratitude to the reviewers for their constructive and useful remarks. Their comments allowed us to identify areas in my manuscript that needed modification.
We also thank you for allowing us to resubmit a revised copy of the manuscript.
We hope that the revised manuscript is now suitable for publication in Biomedicines.
The manuscript is original, and it has not been published or accepted for publication, either in whole or in part, in any form. No part of the manuscript is currently under consideration for publication elsewhere.
Yours sincerely,
Beata Jabłońska, PhD MD
Department of Digestive Tract,
Faculty of Medical Sciences in Katowice,
Medical University of Silesia
Responses to comments:
Comment:
1) The study is retrospective and the sample size is limited. I am aware we are speaking about a rare disease but this is why multicenter series are needed in this field...
Answer:
Thank you for your valuable comment. We agree with your opinion. Therefore, we have pointed in discussion limitations of our study as follows:
This study has some limitations. First, it was a single centre experience with a group of 22 patients. Moreover it was a retrospective study of a rare condition. A prospective study should be conducted to validate the results in multiple centres.
Comment:
2) The KM is misleading as it appears as some patients died during the follow-up (they are just censored). I would remove it because it adds very little information to the paper.
Answer:
Thank you for your valuable comment. We have removed it according to your suggestions.
Comment:
3) Why only half od patients underwent EUS-tisse acquisition before surgery? Don't you need a definitive diagnosis before surgery??
Answer:
Thank you for your valuable comment. According to European guidelines [European Study Group on Cystic Tumours of the Pancreas. European evidence-based guidelines on pancreatic cystic neoplasms. Gut. 2018 May;67(5):789-804. doi: 10.1136/gutjnl-2018-316027. Epub 2018 Mar 24. PMID: 29574408; PMCID: PMC5890653], EUS is recommended as additional to other radiological investigations. It is helpful for diagnosing pancreatic cystic neoplasm (PCN) indicated for surgery. Similar to MRI and CT, EUS is not perfect in diagnosis of the exact PCN type of EUS is recommended in patients with PCNs with concern clinical or radiological features.
According to European guidelines, EUS-FNA should only be performed when the results are expected to change clinical management (GRADE 2C, strong agreement).
According to European guidelines, EUS-FNA should not be performed if the diagnosis is already established by cross-sectional imaging, or where there is a clear indication for surgery.
In our department, EUS-tissue acquisition was performed in patients in whom the PCN type was uncertain in terms of indications for surgery (i.e. differential diagnosis between lesion not requiring surgery such as SCN and lesion requiring surgery such as SPN/MCN/small IMPN). In patients, who were qualified for surgery based on CT/MRI, EUS was not performed. For example, preoperative differentiation between SPN and MCN of diameter 4 cm is not necessary, because both tumors should be removed and the type of surgery depends on PCN location.
Comment:
4) Why the patients underwent EUS-FNA instead of FNB which performs much better? In fact, most patients had incorrect or indefinite diagnosis......The authors should also comment in the Discussion that EUS-FNB is the gold standard for the diagnosis before surgery in these patients (in this regard, cite two relevant papers on the impact of FNB in solid pancreatic lesions including SPN: PMID: 36657607 and PMID: 35915956).
Answer:
Thank you for your valuable comment. We apologize for this misunderstanding. In fact, in our patients, EUS-FNB was performed, because in all of them, representative specimen for histopathological investigation was taken. According to European guidelines, EUS-FNA improves diagnostic accuracy in PCN for differentiating mucinous versus non-mucinous PCN, and malignant versus benign PCN, in cases where CT or MRI are unclear. In our manuscript the term used in European guidelines (EUS-FNA) was presented, but according to your suggestion we have corrected it in order to precise this diagnostic tool. In discussion, we have written about biopsy as follows:
Therefore fine-needle aspiration biopsy (FNA) may be useful in obtaining diagnosis [11,29].
In addition, according to your suggestion, we have cited two relevant papers on the impact of FNB in solid pancreatic lesions including SPN: PMID: 36657607 and PMID: 35915956) as follows:
However, FNB is considered as a gold standard for the diagnosis of solid pancreatic lesions including SPN before surgery because of obtaining the histologic core tissue specimen while using a biopsy needle and the wet-suction technique resulted in higher acquisition rate in comparison to the slow-pull method with similar diagnostic accuracy between those two methods [31, 32].
Reviewer 2 Report
Comments and Suggestions for Authors
I have to congratulate the authors for this interesting and original work performed.
The authors report about “Pancreatic solid pseudopapillary neoplasms – clinicopathological characteristics and surgical outcomes: a 10-year single-center observational study”.
It conveys valuable data from a relatively rare tumor entity, with specific attention to clinicopathological features and surgical outcomes.
Nevertheless, I would like to make some comments about the paper presentation.
My comments are the following:
INTRODUCTION
The introduction provides a broad and relevant overview of solid pseudopapillary neoplasms (SPNs) of the pancreas. It touches on key aspects such as epidemiology, clinical presentation, pathology, diagnosis, and treatment.
The text addresses essential elements of SPNs: classification, prevalence, demographics, clinical features, histopathology, and treatment.
METHODS
The “materials and methods” section is well done, it specific and highlights the salient aspects.
RESULTS
-Please put all tables and figures at the end of the manuscript, below the references. Check the journal's guidelines for authors.
- The data are valuable, especially given the rarity of this tumor. However, data interpretation could be improved to enhance readability and scientific rigor.
The results are clearly divided into logical subsections (e.g., general characteristics, symptoms, imaging, diagnosis, surgery, complications), which improves navigability.
|
Reduce overuse of percentages; prefer raw numbers in small cohorts. |
DISCUSSION
-It should highlight that the sample size of this study, although interesting and useful given that it is a rare tumor, is very small.
CONCLUSION
-Please check and improve the English of the manuscript
Comments on the Quality of English Language-Please check and improve the English of the manuscript
Author Response
Dear Reviewer,
This is the revised original article „Pancreatic solid pseudopapillary neoplasms – clinicopathological characteristics and surgical outcomes: a 10-year single-center observational study ” (biomedicines-3782585) for the special issue "Pancreatic, Liver, Biliary Tract and Intestinal Diseases: Pathogenesis, Diagnostics and Therapy—2nd Edition" of the “Biomedicines” journal.
Thank you for your questions and comments. We have fully addressed all the comments and our responses appear below. Our revised work includes corrections according to reviewers’ comments in the text. Our revisions, made according to reviewers’ comments, are marked using the red font function in the manuscript.
We take this opportunity to express our gratitude to the reviewers for their constructive and useful remarks. Their comments allowed us to identify areas in my manuscript that needed modification.
We also thank you for allowing us to resubmit a revised copy of the manuscript.
We hope that the revised manuscript is now suitable for publication in Biomedicines.
The manuscript is original, and it has not been published or accepted for publication, either in whole or in part, in any form. No part of the manuscript is currently under consideration for publication elsewhere.
Yours sincerely,
Beata Jabłońska, PhD MD
Department of Digestive Tract,
Faculty of Medical Sciences in Katowice,
Medical University of Silesia
Responses to comments:
Comment: INTRODUCTION
The introduction provides a broad and relevant overview of solid pseudopapillary neoplasms (SPNs) of the pancreas. It touches on key aspects such as epidemiology, clinical presentation, pathology, diagnosis, and treatment.
The text addresses essential elements of SPNs: classification, prevalence, demographics, clinical features, histopathology, and treatment.
Answer:
Thank you for your positive feedback.
Comment: METHODS
The "materials and methods" section is well done, it specific and highlights the salient aspects.
Answer:
Thank you for your positive feedback and valuable comment.
Comment: RESULTS
-Please put all tables and figures at the end of the manuscript, below the references. Check the journal's guidelines for authors.
- The data are valuable, especially given the rarity of this tumor.
However, data interpretation could be improved to enhance readability and scientific rigor.
The results are clearly divided into logical subsections (e.g., general characteristics, symptoms, imaging, diagnosis, surgery, complications), which improves navigability.
Reduce overuse of percentages; prefer raw numbers in small cohorts.
Answer
Thank you for your positive feedback and valuable comments. We have checked the guidelines for authors and according to them tables and figures should be used in the main text close to their citation. Here is what we have found on the page of Biomedicines in article named: Instructions for Authors in Preparing Figures, Schemes and Tables section as follows:
“Order:
- All figures, schemes and tables should be inserted into the main text close to their first citation and must be numbered following their order of appearance (e.g., Figure 1, Scheme 1, Figure 2, Scheme 2, Table 1, etc.).”
According to your suggestion we have resigned from percentages and have used raw numbers.
Comment: DISCUSSION
-It should highlight that the sample size of this study, although interesting and useful given that it is a rare tumor, is very small.
Answer:
Thank you for your valuable comment. We agree with your opinion. Therefore, we have pointed in discussion limitations of our study as follows:
This study has some limitations. First, it was a single centre experience with a group of 22 patients. Moreover it was a retrospective study of a rare condition. A prospective study should be conducted to validate the results in multiple centres.
According to your suggestion we have added the sentence about small group of patients in the beginning of discussion as follows:
In the conducted small group study only 1 of 22 patients experienced recurrence and needed another operation.
Comment: CONCLUSION
-Please check and improve the English of manuscrpit
Answer:
Thank you for your valuable comment. We have checked and revised the English of the manuscript.
Round 2
Reviewer 1 Report
Comments and Suggestions for Authors
The revised manuscript is OK